# Contrast-Enhanced CT Renal Tumor Segmentation

ChuDa Xiao[1], Haseeb Hassan[1,2], Bingding Huang[1*]

[1]College of Big data and Internet, Shenzhen Technology University, Shenzhen, China.
[2]Guangdong Key Laboratory for Biomedical Measurements and Ultrasound Imaging, School of Biomedical Engineering, Shenzhen University Health Science Center, Shenzhen, China.
*Corresponding author: Bingding Huang (huangbingding@sztu.edu.cn)

**Abstract.** Automated detection and segmentation of kidneys, tumors, and cysts are useful for renal diagnosis and treatment planning. Here we propose a two-stage contrast-enhanced CT detection and segmentation framework that automatically segments the kidney, kidney tumor, and cyst. Testing the proposed algorithm on the KiTS21 dataset, we achieve the mean dice of 0.6543 and the mean surface dice of 0.4658.

**Keywords:** two-stage; kidney tumor; segmentation

## 1 Introduction

Renal refers to the kidneys. The terms "tumor" and "mass" refer to abnormal body growths. Our kidneys might develop masses (growths or tumors) from time to time. Some kidney tumors are benign (noncancerous), whereas others are malignant (cancerous). According to GLOBOCAN data from 2018, an estimated 403,000 persons are diagnosed with abnormal kidney growth each year, accounting for 2.2% percent of all cancer diagnoses [1]. Due to the enormous variation in kidney and kidney tumor shape, there is a lot of interest in understanding how tumor morphology influences surgical outcomes [2, 3] and developing sophisticated surgical planning techniques [4].

For this purpose, measuring the shape and dimensions of a kidney tumor can be revealed by contrast-enhanced Computed Tomography (CT) imaging which is essential for diagnosis, treatment, and safe surgery [5]. Safe surgery involves avoiding injury to the kidney's vascular network. As a result, automatic semantic segmentation becomes a critical component of surgical planning and is widely used. Previously the KiTS2019 [6] focused on kidney and kidney tumor segmentation, whereas the newly introduced challenge KiTS21 includes an additional class of cyst. Therefore, in this article, we propose to segment kidney, kidney tumor, and cyst. This remaining manuscript is organized as follows. Methods and procedures are defined in section 2, section 3 presents the experimental results, and section 4 concludes the overall manuscript.

## 2 Methods

In this work, we use a multi-stage algorithm to segment kidney, tumor and cyst. The proposed algorithm comprises two stages, as depicted in Figure 1. The first stage involves the detection process, and the second stage applies the segmentation process. In both settings, we consider ResUnet 3D as the backbone network. The detection process we use for accurate localization of the kidney. The reason for this, to make the subsequent segmentation process more effective.

Primarily, we preprocess the CT training data by resampling (the z-axis spacing to 2mm, while retaining the x, y-axis unchanged) and cropping them to sizes 32x384x384. These CT images are provided as an input to the detection network. The detection network initially detects both the kidneys' shapes based on the preprocessed inputs. After the detection process, we calculate the centers of each detected kidney according to (x, y, z) points by using the *skimage* library. Further, the detected kidneys were cropped into cube (volume) sizes 64x128x128 and provided that to the segmentation network to predict kidney, tumor, and cyst. After that, we combine all the predicted masks from the segmentation network to the volume size ($s \times 384 \times 384$). Finally, we post-process (by padding and resampling) the combined volume size ($s \times 384 \times 384$) to the original CT scan size, i.e. ($n \times 512 \times 512$).

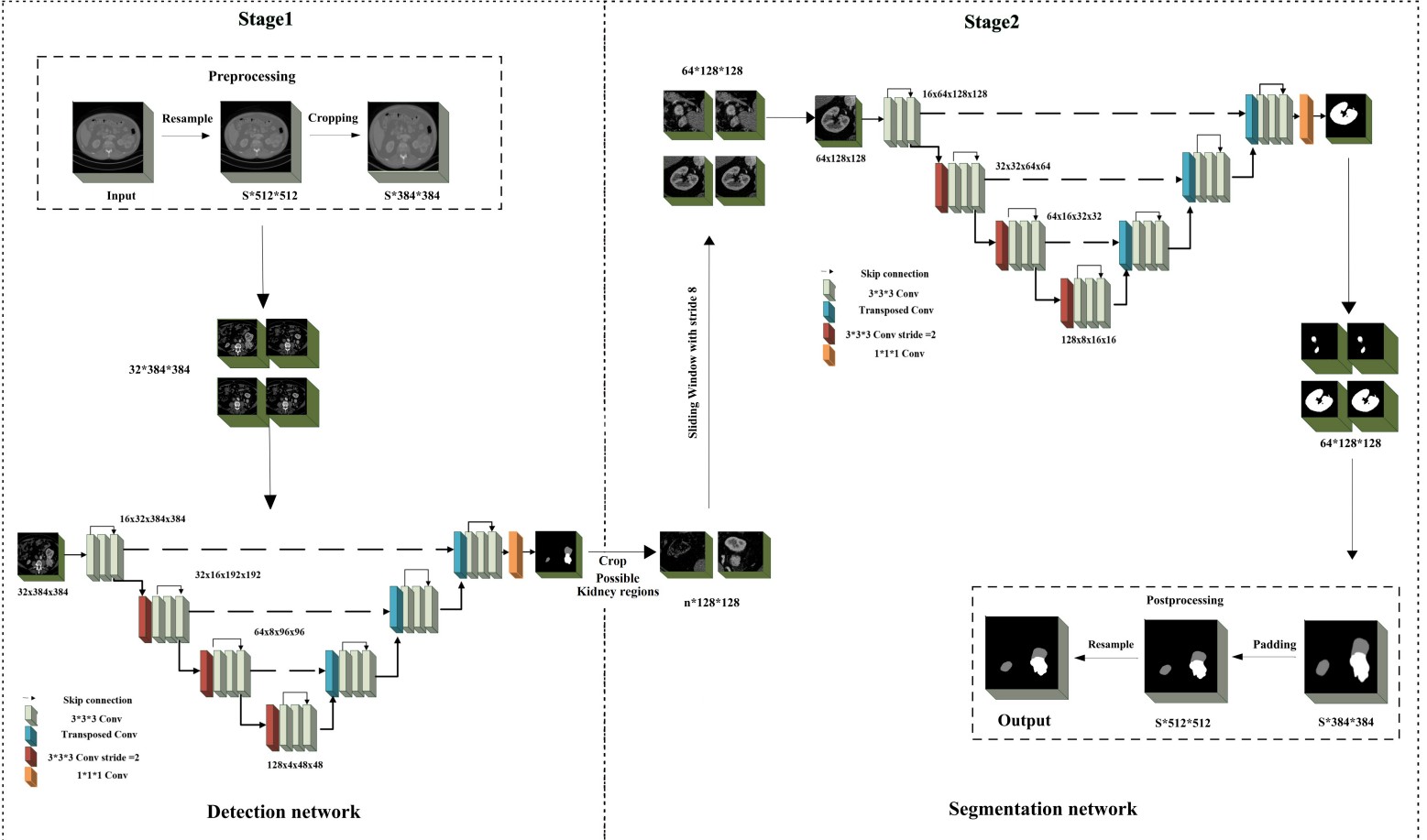

Figure 1: The architecture of our proposed two-stage detection and segmentation framework.

## 2.1 Training and Validation Data

To validate our proposed framework, we use the official KiTS21 dataset. The dataset is divided into training and validation sets (training: 269 cases, validation: 30 cases). For instance, 00000 to 00269 are training cases, and 00270 to 00299 are validation cases. Note that we removed case 00160 from the adopted dataset due to its different size. In addition, we use voxel-wise majority voting (MAJ) for training and validation.

## 2.2 Preprocessing

After analyzing the KiTS21 data, we found different z-axis spacing between cases, i.e., in the range [0.5mm-5mm]. In the dataset, around 125 cases where the z-axis spacing is 5mm. To balance the z-axis spacing between each case, we resample the data to size ($s$, 512, 512) and set the z-axis spacing to 2mm, while the x-axis and y-axis remain unchanged.

Further, we discard the useless regions from each resampled CT slice and crop it to size ($s$,384, 384) according to the image's center point for both training and prediction. Note that $s$ denotes the number of resampled slices. The reasons for cropping are to reduce the image size and to cover the possible kidney regions. To improve the GPU utilization, we use the sliding window with a stride of 8 to resize the CT images to the volume size (32, 384, 384). We also perform some data cleaning by removing those volumes not containing the kidney, tumor, and cyst.

After the kidneys detection in stage 1, we separate left and right kidney regions according to their masks. To cover the entire kidney in both left and right regions, we crop the kidney to volume sizes (64,128,128) with a stride of 8 by the following formula.

$$Counter_{volume} = \frac{n-64}{8} + 1 \qquad\qquad \text{Eq. (1)}$$

In the above Eq. 1, n is the number of slices.

Meanwhile, to improve the network accuracy, the intensity of the other organs of the CT is reduced by normalizing the HU intensity to the range [-100, 300]. The normalized HU intensity range is further subtracted by 100, i.e., [-200, 200] and divided by 50, which is more useful for CNNs [7] to process.

In addition, we use the data augmentation technique such as horizontal flip, translation, affine translation, etc. For stage 1, the training set size is extended to 18576 volumes, and the validation set is extended to 1914 volumes. Similarly, for stage 2, the training set size is extended to 27066 volumes, and the validation set size is extended to 2550 volumes. Finally, we resample the z-axis spacing of combined CT scans into the original CT scans' z-axis spacing.

### 2.3 Proposed Method

### 2.3.1 Network Architecture

Since U-Net[8] has achieved excellent segmentation results specifically for 3D volumetric CT scans, which are 2D image sequences, therefore, our intended model also takes advantage of U-Net 3D [1] and the Residual network [9] to perform the three-class segmentation task. Our proposed ResU-Net 3D network architecture is shown in Figure 2.

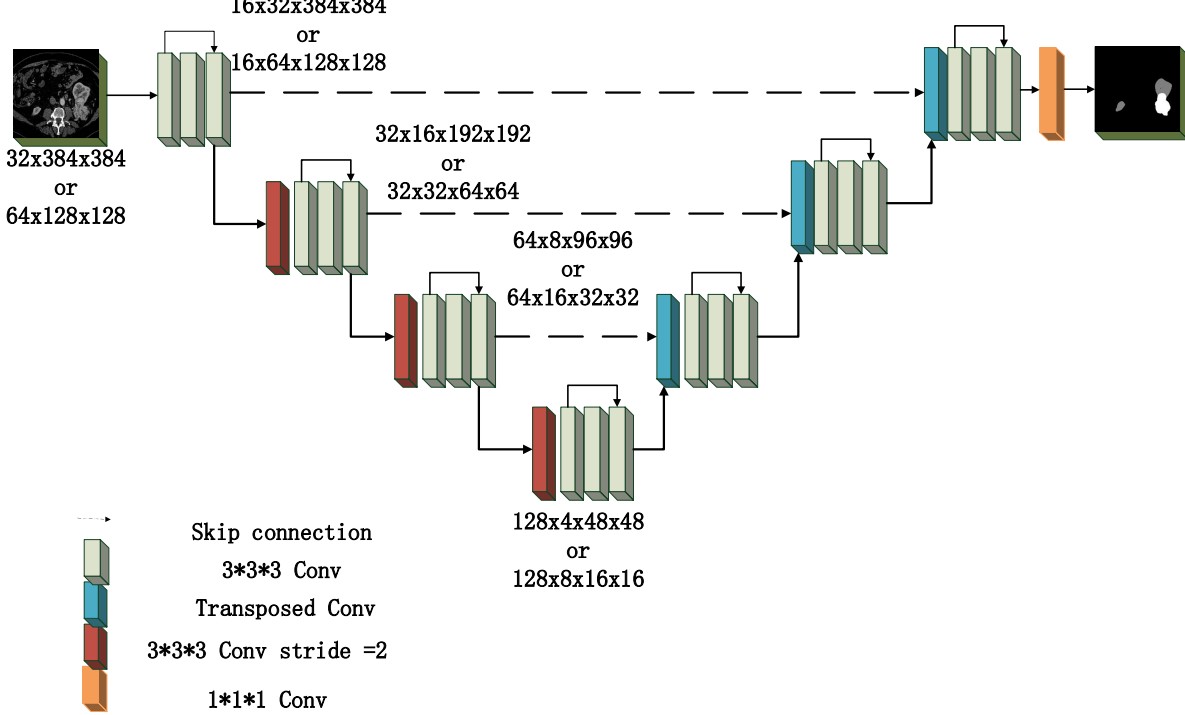

Figure 2：The proposed ResU-Net3D network

The intended network uses three encoder and decoder blocks, which are residual convolutional blocks. The residual block contains three convolutional layers, and the residual convolution kernel size is 3×3×3. The stride size of every residual convolution is 1×1×1. In addition, the up-sampling using the transposed convolution and the up-sampling's convolution kernel size is 2×2×2 where the stride is 2. The down-sampling stride size is 2×2×2 convolution, and the final output layer is 1×1×1 convolution. The network parameters are given in Table 1.

Table 1: The network parameters

| Name | Layers | Stride | Kernel size | Padding |
|---|---|---|---|---|
| Convolution(C) | Conv | - | - | |
| | Batch norm | - | - | |
| | ReLU | - | - | |
| Residual Block | C1 | 1×1×1 | 3×3×3 | 1×1×1 |
| | C2 | 1×1×1 | 3×3×3 | 1×1×1 |
| | C3 | 1×1×1 | 1×1×1 | 0 |
| | C1+C3 | | | |
| Down Sample | C | 2×2×2 | 2×2×2 | 0 |
| Up Sample | Transpose Conv | 2×2×2 | 2×2×2 | 0 |
| Concat | Residual Block + Up Sample | | | |
| Output Layer | C | 1×1×1 | 1×1×1 | 0 |

### 2.3.2 Training

Our proposed networks are implemented using the Pytorch1.9.0 framework. To train the network, we use the Adam optimizer as the network optimizer. The initial learning rate is set to 0.001, and we choose cross-entropy to calculate the loss function of the network. For stage 1, the input volume sizes are 32×384×384, and for stage 2, the input volume sizes are 64×128×128. The batch size in stage 1 is set to 4 and 16 in stage 2. A total of 50 iterations (epochs) are performed for training the network on 32G Nvidia V100 GPU.

### 2.3.3 Validation strategy

According to Hierarchical Evaluation Classes (HECs) proposed by the KiTS21 challenge, the following HECs will be used.

- Kidney and Masses: Kidney + Tumor + Cyst
- Kidney Mass: Tumor + Cyst
- Tumor: Tumor only

To evaluate the performance of the model, we use dice and surface dice(SD).

## 3  Results

We selected case00270~case00299 as the validation set and provided these cases' to the proposed pipeline for prediction purposes. Table 1 shows the three classes dice and surface dice where KMC denotes kidney and Masses class, Kidney Mass is denoted by KM, and Tumor class is denoted by T. Table 2 shows the achieved Mean Dice and surface dice are 0.6543 and 0.4658. Figure 4 provides the visual analysis of the proposed pipeline prediction.

Table 1. Mean Dice and SDS of classes on KiTS21 validation set

| Network | KMC Dice | KM Dice | T Dice | KMC SD | KM SD | T SD |
|---|---|---|---|---|---|---|

| **Ours** | 0.9130 | 0.5635 | 0.4864 | 0.7718 | 0.3424 | 0.2834 |

Table 2. Mean Dice of the proposed pipeline on KiTS21 validation set

| Network | Mean Dice | Mean SD |
|---------|-----------|---------|
| **Ours** | 0. 6543 | 0. 4658 |

Figure 4：Visualization of predictions of our proposed model. The 1st column is the input CT slice, the 2nd column is the mask, and the third column is our proposed method predictions.

## 3  Discussion and Conclusion

This work proposed two-stages detection and segmentation architecture to automatically segment kidney, cyst, and tumor based on the KiTS21 benchmark. For both the detection and segmentation networks, the ResUnet 3D is utilized as the backbone. The designed two-stage architecture achieved the mean dice of kidney and messes, kidney messes and the tumor is 0.6543, and the mean surface dice is 0.4658. However, our model generated low dice and surface dice for the tumor and cyst. The reason for that is that tumor and cyst are quite tiny and have limited availability in the adopted dataset. Therefore, to address this problem, in the future, we plan to augment data of the tumor and cyst for better detection and segmentation, which will eventually lead us to better quantitative outcomes.

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

**Response to Reviewers' comments**

We are highly thankful to the KiTS21 organizing committee and reviewers for putting their efforts and time into our manuscript peer review process. We are also grateful to reviewers for their insightful analysis and valuable suggestions regarding our work. We revised the whole manuscript and made the required changes recommended by the reviewers. The modified changes are highlighted in the manuscript. The point-to-point responses to the reviewers' comments are provided as follows.

## Review 1:
### Overall
- Please make sure to update the abstract with your performance on the test set once this is known

### Introduction
- Looks good

### Methods

- It might be nice to mention the resampling earlier, perhaps before you mention cropping
  Response: Thank you for your useful suggestions. We modified our manuscript according to the suggested guidelines.

- Am I understanding correctly that you simply discard all pixels outside of a 384x384 window? Is this only during training? Please clarify within the paper

  Response: Yes, we discard all the outside pixels in both training and prediction. We also clearly mentioned this in the revised version of our manuscript.

- Please explain why you chose your HU bounds for normalization

  Response: Thank you for your deep analysis. Choosing HU bounds normalization is explained as follows.
  "Meanwhile, to improve the network accuracy, the intensity of the other organs of the CT is reduced by normalizing the HU intensity to the range [-100, 300]. The normalized HU intensity range is further subtracted by 100, i.e., [-200, 200] and divided by 50, which is more useful for CNNs [7] to process." We also clearly mentioned this in the modified version of our manuscript.

### Results
- Looks good

### Discussion and Conclusion
- Looks good

**Rating:** 7: Good paper, accept

## Review 2:

This paper presents a coarse-to-fine approach based on a 3D U-Net. The paper is well-written and contains a good level of detail along with nice figures. One point that was omitted, though, is which set of aggregated segmentations they used for training and validation? Most teams used majority voting, but some used the sampling method from the GitHub repo. The authors should include this detail in their revision.

  Response: Thank you for your insightful analysis. We use voxel-wise majority voting (MAJ) for training and validation. We also indicated this in the revised version of our manuscript.

**Rating:** 7: Good paper, accept