# OpenReview forum: "Contrast-Enhanced CT Renal Tumor Segmentation"
_MICCAI.org/2021/Challenge/KiTS — Submitted to KiTS21 Challenge_

### Official Review · Reviewer_pdBA · 2021-08-30

**Rating:** 7

**Review:**

This paper presents a coarse-to-fine approach based on a 3D U-Net. The paper is well-written and contains a good level of detail along with nice figures. One point that was omitted, though, is which set of aggregated segmentations they used for training and validation? Most teams used majority voting, but some used the sampling method from the GitHub repo. The authors should include this detail in their revision.

---

### Official Review · Reviewer_AFHc · 2021-08-30

**Rating:** 7

**Review:**

### Overall

- Please make sure to update the abstract with your performance on the test set once this is known

### Introduction

- Looks good

### Methods

- It might be nice to mention the resampling earlier, perhaps before you mention cropping
- Am I understanding correctly that you simply discard all pixels outside of a 384x384 window? Is this only during training? Please clarify within the paper
- Please explain why you chose your HU bounds for normalization

### Results

- Looks good

### Discussion and Conclusion

- Looks good

---

### Decision · Program_Chairs · 2021-08-30

**Decision:**

Minor Revisions

**Comment:**

Please address the reviewer comments and resubmit